# Aerogeophysical characterization of an active subglacial lake system in the David Glacier catchment, Antarctica

Laura E. Lindzey[1,2,a], Lucas H. Beem[3], Duncan A. Young[1], Enrica Quartini[1,2,b], Donald
D. Blankenship[1,3], Choon-Ki Lee[4], Won Sang Lee[4], Jong Ik Lee[5], and Joohan Lee[6]

[1]University of Texas Institute for Geophysics, University of Texas at Austin, Austin, TX, USA
[2]Department of Geological Sciences, Jackson School of Geosciences, University of Texas at Austin, Austin, TX, USA
[3]Montana State University, Bozeman, Montana, USA
[4]Unit of Ice Sheet and Sea Level Changes, Korea Polar Research Institute, Incheon 21990, South Korea
[5]Division of Polar Earth-System Sciences, Korea Polar Research Institute, Incheon, 21990, South Korea
[6]Division of Polar Technology, Korea Polar Research Institute, Incheon 21990, South Korea
[a]now at Department of Ocean Engineering, University of Washington Applied Physics Laboratory, Seattle, WA, USA
[b]now at Georgia Institute of Technology, Atlanta, Georgia, USA

**Correspondence:** L. E. Lindzey (lindzey@uw.edu)

**Abstract.** In the 2016-2017 austral summer, the University of Texas Institute for Geophysics (UTIG) and the Korea Polar Research Institute (KOPRI) collaborated to perform a helicopter-based radar and laser altimeter survey of lower David Glacier with the goals of characterizing the subglacial water distribution that supports a system of active subglacial lakes and informing the site selection for a potential subglacial access drilling project. This survey overlaps with and expands upon an earlier survey of the Drygalski Ice Tongue and the David Glacier grounding zone from 2011 and 2012 to create a 5 km resolution survey extending 200 km upstream from the grounding zone. The surveyed region covers two active subglacial lakes and includes re-flights of ICESat ground tracks that extend the surface elevation record in the region. This is one of the most extensive aerogeophysical surveys of an active lake system and provides higher resolution boundary conditions and basal characterizations that will enable process studies of these features. This paper introduces a new helicopter-mounted ice-penetrating radar and laser altimetry system; notes a discrepancy between the original surface-elevation-derived lake outlines and locations of possible water collection based on basal geometry and hydraulic potential; and presents radar-based observations of basal conditions that are inconsistent with large collections of ponded water, despite laser altimetry showing that the hypothesized active lakes are at a high-stand.

## 1 Introduction

David Glacier is a large East Antarctic outlet glacier, draining $\sim 4\%$ of the East Antarctic Ice Sheet (Rignot, 2002) through the Transantarctic Mountains and into the Western Ross Sea. Smith et al. (2009) identified six active lakes in the David Glacier

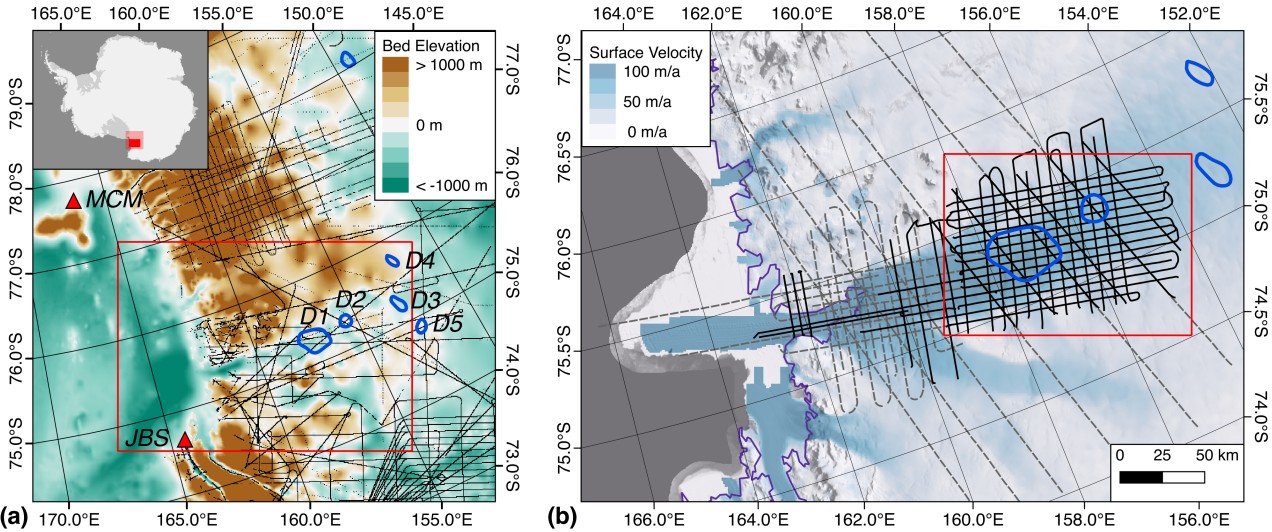

**Figure 1. (a)** Context of David Glacier, showing Bedmap2 bed elevations and radar echo sounding (RES) coverage (Fretwell et al., 2013) prior to the survey reported here. The ASAID grounding line is shown in purple (Bindschadler et al., 2011), and active lakes are outlined in blue and labeled according to Smith et al. (2009). Locations of Jang Bogo (JSB) and McMurdo (MCM) stations are shown with red triangles, and the red box indicates the extent of panel (b). Inset shows locations of panels (a) and (b). **(b)** Locations of post-Bedmap2 RES data. Dashed gray lines were flown by ICECAP in the 2010 and 2011 seasons (ICP3, ICP4). The black lines are flights from the 2016 KOPRI season. Only the 2016 data was used in this work. Background is surface velocity (Rignot et al., 2011a) over the MODIS mosaic (Scambos et al., 2007). The red box indicates the extent of Figures 4, 6 and 8-11. This figure was created using QGIS (QGIS_Development_Team, 2018) and Quantarctica (Matsuoka et al., 2018).

catchment, and their location near KOPRI's Jang Bogo Station makes them an attractive target for a detailed geophysical study. Hypothesized outlines for the lakes are shown in Fig. 1, and they are numbered ascending with distance from the grounding zone, with D1 the farthest downstream. Over the ICESat period (2003-2009), lakes D1, D4, D5 and D6 were observed to be filling, while both D2 and D3 were observed to be draining. A more recent analysis using CryoSat-2 data compared the patterns of surface elevation change within and outside the D1 and D2 lake polygons and concluded that these might not be true lake features because the surface elevation changes were small and did not have a phase difference across the nominal lake boundaries (Siegfried and Fricker, 2018).

After their first identification using RADARSAT InSAR in the Siple Coast (Gray et al., 2005), individual active lakes were observed by orbital remote sensing at locations across Antarctica, including Adventure Subglacial Trench (Wingham et al., 2006), Whillans Ice Stream (Fricker et al., 2007), and Byrd Glacier (Stearns et al., 2008). This was followed by a continent-wide inventory based on the ICESat surface elevation time series (Smith et al., 2009). These surface features were initially hypothesized to reflect the motion of water at the bed based on their monopole nature and the isolated areas of elevation change that occur in consistent locations over time. In further support of a hydrological origin of these features, potential water

routing and even volume balance has been established for the Adventure Subglacial Trench lakes (Wingham et al., 2006; Carter et al., 2009). In addition, the WISSARD project found a thin cavity (~1 m) of water (Tulaczyk et al., 2014) after drilling into Subglacial Lake Whillans. Presumed active lake drainage events have also been associated with ice acceleration in the Byrd Glacier (Stearns et al., 2008), Whillans/Mercer Ice Stream (Siegfried et al., 2016), and Thwaites Glacier (Smith et al., 2017)

catchments, which would be consistent with hydrologically induced modification of the basal boundary conditions.

In addition to occurring in a range of geological settings, active lake drainage events have been observed as a response to both climatic and internal forcings. As an example of the former (and an outlier among active lake observations), Scambos et al. (2011) identified a subglacial lake drainage event apparently triggered by the lowering of Crane Glacier subsequent to the collapse of the Larsen B Ice Shelf. Many of these features have been observed to be cyclical, indicating that they are not all

one-time events triggered by crossing a physical threshold due to changing driving forces, and instead are an ongoing feature of Antarctic hydrology occurring independently of changes to ice sheet geometry. In this view, active lakes are features of the subglacial hydrologic systems representing a stable limit cycle that naturally occurs whenever total melt produced by shear margin heating, basal friction, and/or geothermal flux lies in a critical range too high to be drained via distributed flow and too low to keep channels open. This mechanism appears in the model developed by Werder et al. (2013) and is applied to an

idealized Antarctic system by Dow et al. (2016).

Other work has used the lake outlines from Smith et al. (2009) to constrain ice sheet models. First, it is reasonable to assume that the region of an active lake is at the pressure melting point, which in turn can be a constraint on geothermal heat flux (Van Liefferinge et al., 2018; Pattyn, 2010). Lake boundaries are also used to assume regions of zero basal shear stress (e.g Pattyn, 2010; Matsuoka et al., 2012) and only considered horizontal stress gradients. These usages are dependent on the assumption

that active lake boundaries exactly correspond to the extent of the surface expression and represent ponded water independent of lake stage.

Better understanding the potential link between active lakes and ice dynamics requires better characterization of subglacial water organization, ideally with an observation of basal conditions that is also associated with ice dynamics. A number of ice-penetrating radar surveys have traversed or flown over active lake sites, and significant differences exist between the radar

signature observed at active lakes and established radar lakes (e.g. Siegert et al., 1996; Wright et al., 2012).

Subglacial lakes have long been identified as bright, specular locations in radargrams that are also hydraulically flat and at a local hydraulic potential minimum (Oswald and Robin, 1973; Siegert et al., 1996; Carter et al., 2007; Wright et al., 2012). Some of these locations have been shown to correspond to deep, stable lakes (Kapitsa et al., 1996). The criteria have varied slightly based on the author and data used, and radar-based requirements include relative or absolute brightness based on reflection

coefficient analysis, and smoothness, as inferred from lack of fading (Carter et al., 2007) or high specularity content (Young et al., 2017). Some subglacial lakes have been hypothesized based on their ice surface expression alone (Jamieson et al., 2016).

There have been a number of radar studies of active lake regions (Welch et al., 2009; Langley et al., 2011; Christianson et al., 2012; Wright et al., 2012; Siegert et al., 2014). Most attempt to apply the Carter et al. (2007) lake-detection criteria to the basal horizon under the lake outline proposed by Smith et al. (2009), which usually fails to result in a "definite" lake. There

have been a handful of exceptions to this, where newly-discovered active lakes previously appeared in a radar lake inventory.

Subglacial Lake Mercer appeared in Carter et al. (2007)'s inventory as a definite lake (Fricker and Scambos, 2009), though the rest of the Siple Coast active lakes did not. Additionally, the recipient lakes of the Adventure Subglacial Trench flood were in an existing inventory (Wingham et al., 2006; Siegert et al., 2005), and there are "fuzzy lakes" (lakes lacking a coherent reflection, (Carter et al., 2007)) along its flow path (Carter et al., 2009).

More commonly, these investigations have found a minimum in the basal hydraulic potential and a region of elevated reflection coefficient corresponding with the surface feature. For example, using a survey with multiple ice penetrating radar transects intersecting over a single lake, Siegert et al. (2014) investigated radar characteristics of an active lake in the Institute Ice Stream and observed that the surface elevation signal was associated with an apparently bright (but not smooth or flat) region on the downstream side of a bedrock bump.

Elsewhere, indications of subglacial water have been entirely absent. In the Byrd catchment, Welch et al. (2009) looked at ground penetrating radar data from a traverse, and Wright et al. (2014) used airborne ice penetrating radar to investigate a number of the active lakes identified in the Smith et al. (2009) catalog. None of the locations had clear RES evidence of a water/ice basal interface and Wright et al. (2014) point out that their survey covered a large enough number of lakes that all of them being drained would be unlikely. Langley et al. (2011) attempted similar analysis in the upper Recovery system. Welch

et al. (2009) and Langley et al. (2011) conclude that their observations are consistent with a drained or nonexistent lake, but both fail to compare surface altimetry to the ICESat record to determine whether this is consistent with the surface-elevation derived lake stage.

In this paper, two active subglacial lakes and the surrounding basal environment are surveyed by airborne radar. The survey results show no distinct bed character, in either reflection coefficient or specularity content, beneath the previously established

polygons describing lake extent. Instead the regions shows a high degree of heterogeneity, anisotropy, and surface elevation change inconsistent with their boundaries. These results further confound what the definition of active subglacial lakes should be and how they fit into the broader hydrological system.

## 2   Data

### 2.1   Platform

This paper describes data collected during the 2016-2017 field season using an AS-350 helicopter to fly a UTIG-designed VHF radar (Figure 2). Additional science instrumentation included a Renishaw laser altimeter and a Canon dSLR camera. For precise positioning of the data a Trimble Net-R9 dual frequency carrier phase GNSS and a Novatel SPAN IGM-1A inertial navigation unit were used. After initial test flights, the instruments did not require an operator on board; the pilot had a power switch that was used to disable VHF transmission when necessary. All flights were based out of South Korea's Jang Bogo

Station and supported by the Korea Polar Research Institute (KOPRI).

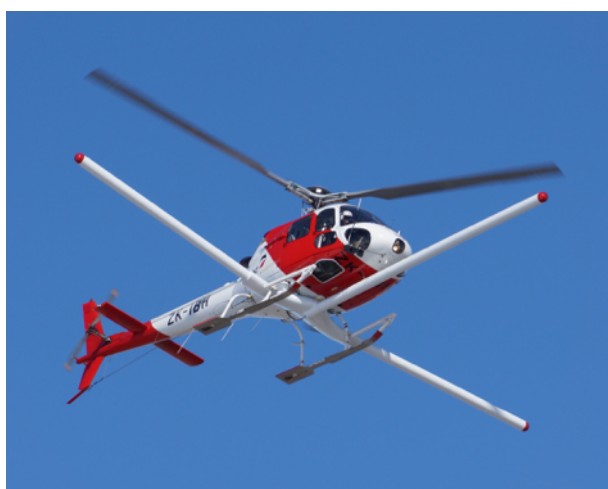

**Figure 2.** Helicopter in survey configuration. Radar antennas were mounted in the lateral booms, which measure 11.52 m tip-to-tip; the front boom was empty. The laser altimeter and dSLR camera were mounted to the right of the pilot's seat, using a pre-existing window.

## 2.2 Laser Altimeter

A Renishaw ILM-1200-HR 905 nm laser altimeter was mounted to the right of the pilot's seat, collocated with the camera, utilizing an existing downward facing window in the helicopter. It provided raw range measurements at 1000 Hz, with 1 cm precision, and its effective max range in Antarctic conditions was $\sim 900$ m. The raw serial stream was recorded by the same acquisition system as the radar data, which provides synchronous timestamps with reference to GPS time.

## 2.3 Ice Penetrating Radar

The new instrument described here is a direct descendant of a lineage of coherent radars that started with an experimental field season in 2001. The original system, termed the High-Capability Airborne Radar Sounder (HiCARS) (Peters et al., 2005), was a hybrid of a JPL-designed coherent radar (Moussessian et al., 2000) and the Technical University of Denmark (TUD) 60 MHz airborne ice penetrating radar system (Skou and Søndergaard, 1976). It was first mounted on a Twin Otter airplane in 2001 to perform surveys of the Siple Coast (Peters et al., 2005), South Pole, and the B15a iceberg (Peters et al., 2007b). This was followed by the 2005 Airborne Geophysical Survey of the Amundsen Sea Embayment, Antarctica (AGASEA) survey of Thwaites, also using Twin Otters (Holt et al., 2006). Since 2008, the International Collaborative Exploration of the Cyrosphere by Airborne Profiling (ICECAP) project has been fielding similar radars using a DC-3T airplane, and UTIG entirely redesigned the electronics with a focus on using commercial, off-the-shelf components to create the HiCARS2 radar in 2010 (Blankenship et al., 2017a, b). In 2014, independent recording from each antenna was added to create the Multifrequency Airborne Radar sounder with Full-phase Assessment (MARFA) (Castelletti et al., 2017), in which digitizer improvements also enabled replacing local oscillator based down conversion with bandpass sampling. The system described in this paper uses the same electronics as MARFA, but with custom antennas for installation on an AS-350 helicopter.

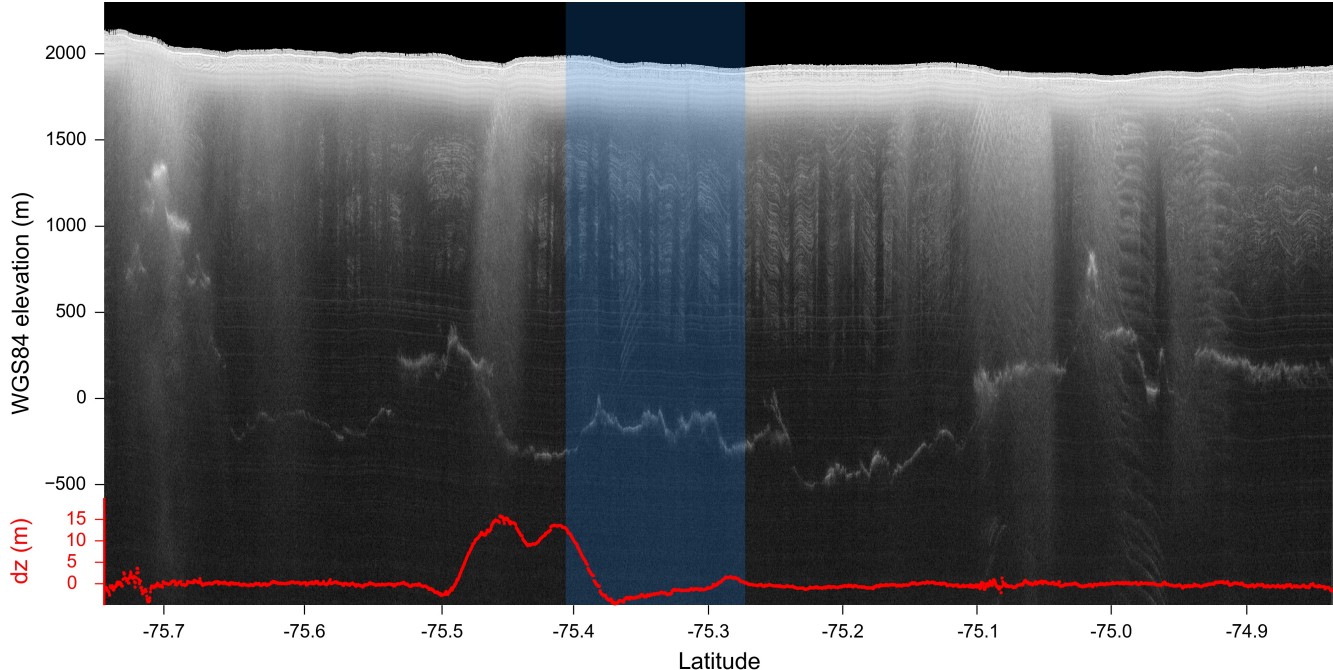

**Figure 3.** Focused radargram DVG/IBH0c/GL0328c. Location of the transect is shown in Fig. 7 and its overlap with the D2 lake outline is highlighted in blue. Observed surface elevation change since 2003 is shown in red. The radargram has been depth-corrected based on laser surface elevations and truncated 20 m above the surface.

The radar transmits 1 $\mu$s wide chirps, linearly sweeping the frequency from 52.7 MHz to 67.5 MHz, with a 6250 Hz pulse repetition frequency and 8 kW peak pulse power. Using separate 14 bit digitizers with low gain for the surface and higher gain for the bed, the system independently records the received signal from each antenna at 50 Msamples/sec, with a total trace length of 3200 samples. The record is stacked 32 times in hardware, then written to disk at 16 bit precision at 196 Hz. This resulted in one raw trace every 18 cm along track at average helicopter ground speed of 70 knots, or 42 cm at the DC-3T ground speed of 160 knots.

The ability to compare data between HiCARS, HiCARS2, MARFA, and the present system has been considered of paramount importance in developing and fielding the new system. The required processing techniques are functionally identical, with differences confined to peak power output/gains and the platform-dependent antennas.

The airplane antennas have a heritage dating back to the 1970's TUD radar. They are center-fed flat-plate dipole antennas suspended $\frac{1}{4}$ wavelength under the wings and mounted inside an airfoil. The helicopter's antennas were designed to fit inside existing flight-certified booms originally designed for magnetometer surveys. These geometric constraints led to an end-fed design with an end plate installed in each lateral boom; the forward boom was empty. The lack of an airplane wing providing a ground plane means that the upward lobe is not reflected, yielding 6 dB lower total system gain. Additionally, the smaller separation between the antennas yields a wider central lobe, leading to increased surface scattering that can be mitigated by

flying closer to the ice surface. There is no evidence in the data for time varying interference due to the helicopter blades rotating at $\sim 400$ rpm.

## 3 Methods

### 3.1 Positioning

Processing of GPS observations were performed using Novatel's Waypoint GrafNav software, which reports $\sim 15$ cm $\sigma$ for precision. All data in this paper are reported with reference to the WGS84 ellipsoid.

### 3.2 Laser Altimeter

All analysis presented in this paper used data that geolocated the median of 100 raw range measurements, which spans $\sim 3.6$ m along-track.

### 3.2.1 Calibration

The laser's mounting bias relative to the INS was solved in a two step process, similar to Young et al. (2008, 2015). The first step used a digital level to obtain a coarse estimate of roll/pitch, but this is insufficient to obtain the desired accuracy in geolocation. In the second step, the measured values are used as the initial seed for a minimization of crossover errors based on data from a dense grid with 150 crossovers flown at three different elevations over a smooth region of the Nansen Ice Shelf.

The resulting calibration used crossover points to compare surface elevations and yielded a standard deviation of 44 cm within that grid. Validation was performed by comparing the new surface elevation estimates to raw ICESat surface elevations where available over slow-moving ice; this revealed no bias in the reported ranges.

### 3.2.2 Surface Elevation

Subglacial lake state at the time of the 2017 survey is determined using two different methods of comparing the new laser

altimetry data to the 2003-2009 ICESat surface elevation record. ICESat's Geoscience Laser Altimeter System (GLAS) measured ice surface elevations at 172 m along track spacing with a $\sim 60$ m radius footprint and 15 cm vertical accuracy (Zwally et al., 2002). It collected data on 91-day repeat orbits with ground tracks separated by $\sim$14-20 km in the David region. Comparison of surface elevation data along repeat tracks is complicated by the fact that the GLAS instrument did not precisely point at the reference track: elevation differences due to cross-track surface slope confound differences due to actual surface

elevation change.

    First, crossovers between the 2017 survey's along-flow lines and all available ICESat data are compared. This is the simplest method of processing surface elevation change, since it compares data at overlapping points and therefore does not require any correction for surface slopes. The resulting elevation change observations are both sparser along the GLAS lines (as determined by the DVG survey spacing) and denser between the nominal GLAS lines because it is possible to include all of

the off-nominal tracks from early in the ICESat era. Across the entire survey, the mean elevation difference is -0.04 m, and the median is -0.20 m, which provides a rough validation of the calibration for the helicopter's laser altimeter.

Next, reflown ICESat tracks are used to compute surface elevation change. This requires adding a correction for cross-track slope since neither the original ICESat orbits nor the reflights exactly sampled the ground track. This paper follows the method

from Smith et al. (2009) to estimate surface slope: perform linear regression to solve for $\frac{dz}{dx}$, $\frac{dz}{dy}$ and $\frac{dz}{dt}$ using all GLAS surface elevation measurements in overlapping windows measuring 700 m along track at 500 m intervals. For each GLAS point, we calculated dz as the vertical distance between that point and the plane that passes through the nearest new observation with the GLAS-based surface slopes. Any GLAS point further than 500 m from the nearest point in the new survey is discarded.

### 3.3 Ice Penetrating Radar

For this work, we used the 1D-focused processing for radargrams described in Peters et al. (2007a) for geometry and basal reflectivity, complemented with 2D focusing to derive specularity content (Schroeder et al., 2015). Figure 3 shows an example radargram that crosses D2. Focusing is performed by convolving a kernel with pulse-compressed radar data, where the kernel is generated based on the expected appearance (delay and phase) of a point scatterer at that location, which is a function of airplane height, ice thickness and surface slope. Different aperture lengths are used for focusing, which correspond to the 1D

and 2D nomenclature in Peters et al. (2007a). 1D focusing uses a short enough aperture that range changes are less than a pulse width; for a longer aperture, a 2D kernel (in this case accommodating 1 $\mu sec$ of range change) is required to match the phase history, further improving resolution, collection of scattered energy and detection of sloping interfaces with some cost to signal to noise ratio.

#### 3.3.1 Topography

For estimating the bed elevation and ice thickness, the radar reflection off the basal interface was identified by a manual process that labels the first returned continuous reflector (Blankenship et al., 2001). The right-side radar antennas had a stall strip that raised its noise floor, so labeling was performed on the data from the left antenna only. (Further processing used the combined product.) Due to the existence of side lobes in the transmit/receive beam pattern, the first return criteria may underestimate ice thickness in rough terrain, overestimate the width of mountains/ridges, and possibly fail to detect valley floors and lakes. Given

the surface and bed horizons, ice thickness is calculated using 168.42 $\frac{m}{\mu sec}$ as the speed of light in ice without a correction for firn density gradient. The bed elevation product results from subtracting this ice thickness from the laser-determined surface elevation.

All intersecting lines where bed picks were recovered within a 100 m radius are used to characterize the uncertainty in bed elevation estimates. There is no attempt to reconcile differing bed estimates from intersecting transects in the labeling process:

picking is purely based on the first-return criteria. This means that the computed crossover differences are valid for projecting to regions without crossovers but with equivalently rough topography. These crossover differences are shown in Figure 4. Of the 450 locations where survey lines intersected and a bed was recovered, 76% had differences less than 50 m; 88% had differences less than 100 m, and 95% were under 200 m. There was no clear spatial pattern to the distribution of errors, and

inspection of the 6 intersections with greater than 400 m difference revealed that the apparent errors were consistent with the observed along-track variation in bed elevation at length scales equivalent to the across-track beam width.

Generation of the Digital Elevation Model (DEM) started with the full-density bed elevation points from every trace at which a bed return was detected. These profiles are preprocessed for gridding by sampling to a 500 m cell size using GMT's (Wessel and Smith, 1998, 1991) `blockmean` function. Natural neighbor interpolation was performed on this decimated data set using matplotlib's (Hunter, 2007) `griddata`, which is based on Delaunay triangulation. This interpolation retained artifacts along the flight lines, so a 1 km standard deviation gaussian filter was applied as a final step. The DEM presented here reports the ice/water interface beneath floating ice: it makes no attempt to mask the grounding line or correct for water column to determine bathymetry.

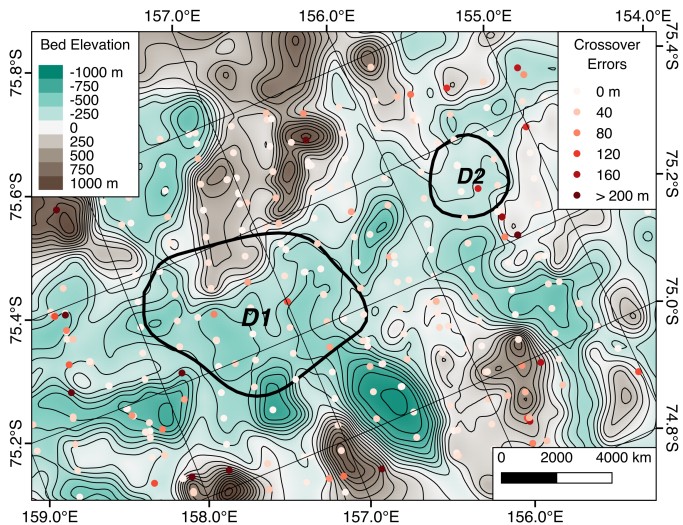

**Figure 4.** Bed elevation DEM generated from KOPRI data, with 100 m contours and errors in bed elevation between intersecting lines. Lake outlines from Smith et al. (2009) are black.

Profile-based ice thicknesses can be problematic to grid due to their anisotropic sampling density. This dataset's line spacing does not support using a higher resolution DEM; therefore, when compared to the raw data, there are sometimes significant gridding errors. They are particularly noticeable in higher-relief areas, where bed features are flattened and broadened. Unlike the crossover errors, these gridding errors follow a roughly normal distribution, with a standard deviation of 95 m. Of the ~half million bed elevation measurements, 53% differ from the gridded product by less than 50 m error, 79% by less than 100 m, and 95% by less than 200 m.

### 3.3.2 Hydraulic Potential

Overall subglacial water flow is largely controlled by hydraulic potential gradients. The organization of the subglacial hydrology is controlled by the geometrical boundary conditions in concert with water production, temperature gradients, and basal

substrate. Remote sensing allows us to characterize the large-scale geometric contributions to hydraulic potential, which is typically expressed as: (Paterson, 1994):

$$\phi = z_{bed}\rho_{water} + h\rho_{ice}, \tag{1}$$

where $\phi$ is subglacial hydraulic potential, $z_{bed}$ is the WGS84 elevation of the ice/bed interface, $h$ is the ice thickness in meters, $\rho_{water} = 1000 \frac{kg}{m^3}$ is the density of fresh water, and $\rho_{ice} = 917 \frac{kg}{m^3}$ is the density of glacial ice.

Equation 1 assumes that subglacial water pressure is at overburden pressure, fully supporting the column of ice above it, and neglects the effects of bridging stresses. Very little data exists for assessing how realistic these assumptions are. Measurements by Engelhardt and Kamb (1997) at the Siple Coast found basal pressures varying within a few percent of overburden. Idealized modeling by Dow et al. (2016) on a simple plane yielded pressure waves ranging from 95 - 104% of overburden pressure. In Greenland, where the basal water system can be connected to the atmosphere via moulins, analysis has used a wider range of subglacial water pressures (e.g. Chu et al. (2016), who considered values as low as 60% of overburden pressures).

This work used laser derived ice surface elevations and radar derived ice thicknesses to calculate hydraulic potential along the profiles. Equation 1 can be refactored to separate the observations of the ice surface elevation and ice thickness:

$$\phi = z_{srf}\rho_{water} - h(\rho_{water} - \rho_{ice}) \tag{2}$$

Since changes in surface elevation have $\sim$9 times the impact on hydropotential gradients as changes to ice thickness, we use laser derived surface elevations, which are more precise than those derived from radar. Profile data was gridded using the same approach as bed elevations.

Following standard propagation of errors for Equation 2, using the $\sigma_h$ from Sect. 3.3.1 and $\sigma_{z_{srf}}$ from Sect. 3.2.1, the uncertainty is estimated as 10 m of hydraulic head. However, this analysis does not include uncertainties due to the assumption that basal water pressure is equal to overburden or the fact that radar observations of bed elevation are likely to entirely miss narrow valleys since the radar itself is more likely to detect a first return from the side before a deeper return from the bed.

### 3.3.3 Reflection Coefficients

The strong dielectric contrast between water and ice means that this reflection should be significantly brighter than one produced by ice and rock. This observation has been used in an attempt to identify subglacial water as early as Robin et al. (1969) and frequently since (e.g. Oswald and Robin (1973); Siegert et al. (1996); Carter et al. (2007)).

The radar equation describes the amplitude of the returned signal at the antenna ($P_r$) in terms of system and environmental parameters (Peters et al., 2005), assuming a specularly reflecting interface:

$$P_r = P_t\left(\frac{\lambda_1}{4\pi}\right)^2 \frac{G_t G_r T_{12}^2 L_{ice}^2}{[2(h+z/n_2)]^2} R_{23}, \tag{3}$$

where $R_{23}$ is the ice/bed reflection coefficient that we are interested in. $T_{12}$ is the air/ice transmission coefficient. Transmitted power ($P_t$), antenna gain due to cross section ($\frac{\lambda_1}{4\pi}$), and the receiver and processing gains ($G_t$, $G_r$) combine to determine the

system gain. The geometric spreading loss:

$$L_s = \left[ \frac{1}{2(h + z/n_2)} \right]^2 \qquad (4)$$

is a function of aircraft height above the ice surface ($h$) and ice thickness ($z$), both of which can be recovered directly from the interpreted radar data, along with the dielectric constant for glacial ice ($n_2 = \sqrt{\epsilon} = 1.78$).

Finally, $L_{ice}$, the energy lost as an electromagnetic wave travels through a dielectric media, is a function of its permittivity. For ice, this depends primarily on temperature and chemistry (Matsuoka, 2011). Across Antarctica, one-way depth-averaged dielectric ice loss ($N_a$) varies from 3 to 30 $\frac{dB}{km}$ (Matsuoka et al., 2012). This wide variation in physically feasible values is the dominant source of uncertainty when calculating reflection coefficients.

Some studies attempt to determine $L_{ice}$ independently from the radar data, either deriving it from first principles based on modeled temperature profiles and salt content (Matsuoka et al., 2012) or extrapolating from measured properties at ice cores (MacGregor et al., 2007). Other studies estimate ice loss from the radar data: Peters et al. (2005) assumed that the brightest echoes correspond to water at the bed and that depth-averaged $L_{ice}$ is constant across the survey area; Jacobel et al. (2009) assumed that the distribution of reflection coefficients is independent of ice thickness, and obtained $L_{ice}$ from the slope of ice thickness vs. geometry-corrected returned power. Recent work has refined these approaches to infer spatially-varying patterns of dielectric ice loss across a survey at a resolution determined by the topographic variation (Schroeder et al., 2016b).

This paper does not attempt to use absolute reflection coefficients to verify the existence of water at the bed. Instead, they are used to compare relative bed properties under similar ice thicknesses. Absolute values would require calibration of the system gain (typically obtained both on the lab bench and by collecting data over open seawater) and validation to previous systems.

Given these goals, we modified the simplest approach of determining depth-averaged dielectric ice loss from the slope of geometry-corrected echo strengths vs. ice thicknesses. This approach relies on the assumption that basal reflection coefficients are independent of ice thickness, which is overly simplifying since basal temperature, and therefore the presence of water at the bed, is correlated with ice thickness. We see evidence of a slope change associated with the likely presence of water, so restricted our linear regression to data in thinner ice. Since thinner ice is on average cooler than thicker ice, restricting the range of thicknesses used in the fit will result in an estimate that is a lower bound on dielectric ice loss.

Figure 5 shows reflection coefficient data derived from peak power measurements from all KOPRI radar bed picks in the study region. In order to address the uneven distribution of samples across ice thicknesses, it also shows the median reflection coefficient for 50 meter bins of ice thickness, which is the input used in calculating the linear regression.

At first look, this does not appear to be a linear distribution – there is a higher slope for the thinner ice than for the thicker ice. It would be surprising if this were due to the distribution of dielectric losses, since thinner ice is typically colder on average, and thus has a lower dielectric ice loss. Instead, this distribution can be explained as a combination of the radar system's noise floor and changing basal properties with depth.

The observed geometry-corrected reflection coefficients have a minimum value of around -110 dB, which serves to cut off the linear distribution. There are areas where the bed can only be identified as a disturbance in the background noise. While the manually interpreted bed picks include these regions, their computed reflection coefficients are not valid. This threshold is

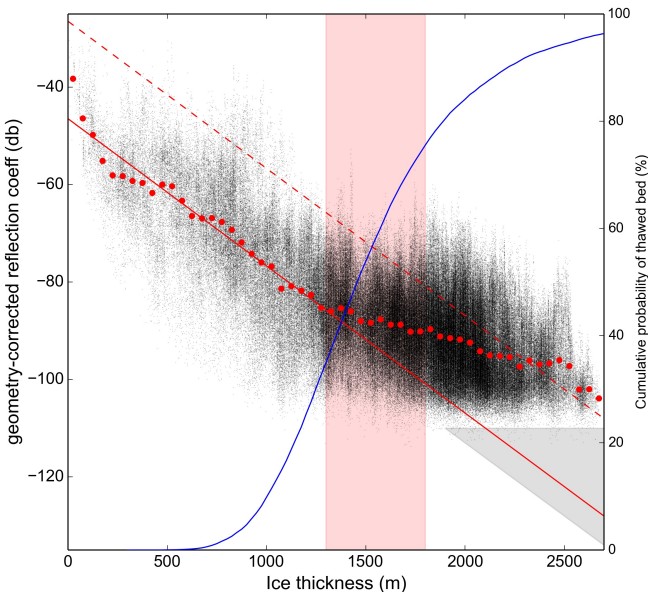

**Figure 5.** Geometry-corrected reflection coefficients vs. ice thickness for all bed picks in the DVG/IBH0c survey. Red dots show the median reflection coefficient for 50 m wide ice thickness bins that were used in calculating the slope. The solid red line shows the slope for -15.1 $\frac{dB}{km}$ (one-way), based on calculations for ice thicknesses less than 1300 m. The red rectangle shows the region where we assume increased presence of water at the bed causes a broadening of the reflection coefficient distribution, and the grey triangle indicates the region where SNR could explain the absence of data. The dotted line has been shifted up 20 dB from the shallow-ice fit, representing the higher reflection coefficients expected if there is water at the bed. The blue line shows the cumulative probability of thawed bed for a given ice thickness.

not a hard limit because the noise distribution varies trace-to-trace. Additionally, the lower bound would be expected to have a slight positive slope due to the effects of correcting for spreading loss, which is apparent in the data.

Liquid water at the bed would be expected to increase the range of observed reflection coefficients, with a maximum value up to 20 dB above those observed on a dry bed. This can explain the observed widening and/or shift of the distribution at depths between 1300 and 1800 m, since the existence of basal water typically requires the insulation provided by thicker ice. Due to the noise floor, a linear fit in this region will underestimate $L_{ice}$. However, the slope of the upper bound of the scatter plot at depths over 1700 m matches the average slope at depths under 1200 m, which supports a roughly constant $L_{ice}$.

Ice thickness required to reach the basal melting point can be estimated using the Robin model (Robin, 1955; Cuffey and Paterson, 2010), which is a 1D model that accounts for ice thickness, accumulation rate, surface temperature, geothermal flux, and basal heat generation. There are many degrees of freedom and in an attempt to simplify and constrain the possible range of solutions a monte carlo approach was adopted. The accumulation rate, in ice equivalent, was assumed to have a normal distribution with one standard deviation of .06 $\pm$ .02 m/a (Van Wessem et al., 2014b). Geothermal flux was assumed to have

a normal distribution of $.06 \pm .01 \frac{W}{m^2}$ (An et al., 2015). Surface temperature is assumed to have a normal distribution of $-35 \pm 5°C$ (Van Wessem et al., 2014a). The standard deviation of the geothermal flux is expected to capture the effects of frictional heating of 0 to $.01 \frac{W}{m^2}$ which is appropriate for up to 50 m/yr of basal sliding (Rignot and Scheuchl, 2017) with 10 kPa of shear stress. Twenty thousand solutions were generated, and the resulting cumulative probabilities of a thawed bed are shown in Figure 5. This analysis indicates that the transition of the bed from predominately frozen to predominately thawed occurs with sufficient degree of likelihood across ice thicknesses consistent with the observed change in basal reflectivity slope.

In combination, these effects can explain the shape of the distribution of observed reflection coefficients vs. ice thicknesses shown in Figure 5. Using a limit of 1300 m, where the basal water is hypothesized to start contributing yields a one-way $L_{ice} = 15.1 \frac{dB}{km}$ ($\sigma = 0.7$), which should be a lower bound within the region.

### 3.3.4 Specularity

Reflection coefficients are problematic for characterizing the basal interface because they do not make it possible to separate the contribution of the dielectric contrast and spatially-varying roughness. Specularity is another property of the radar return that can be informative and is appealing because it is both purely a geometrical property and is dimensionless (the uncertainties introduced in an attempt to calculate absolute reflection coefficients cancel out). Conceptually, it describes how mirror-like a surface is: whether it reflects incident energy directly back or scatters it.

Searching for lakes based on the uniformity and specularity of their signal is not a new concept. It is similar to the old criteria regarding fading, which has been discussed since the initial deployment of ice penetrating radar in Antarctica (Robin et al., 1969). Fading describes how much variation is observed along-track, where uniform surfaces are assumed to vary less than rough surfaces. Other attempts to quantify specularity have involved a proxy looking at the along-track small-wavelength variation in reflection coefficient (Langley et al., 2011; Peters et al., 2005). More recently, Schroeder et al. (2015) defined specularity content ($S_c$) using the ratio between energy captured in different length focusing apertures, building on Peters et al. (2007a)'s observation that different focusing apertures lead to roughness-dependent gains in the focused products. In this analysis, we compute specularity in the same way as Schroeder et al. (2015).

## 4 Results

### 4.1 Lake Stage

Figure 6 shows the spatial distribution of time-normalized elevation changes, as observed using crossings between GLAS lines and the new transects. Figure 7 shows elevation changes computed along individual profiles that followed GLAS ground tracks.

Both the crossovers (Figure 6) and reflight data (Figure 7) are consistent in showing that the downstream part of the D1 outline (as defined by GL0194 and GL0158) has continued to rise, while the upstream portion is inconclusive.

While the region inside the original D2 outline appears to still be lowering with a total displacement of $\sim$5 m, it borders an area along line GL0328 with up to 15 m of elevation *gain* since the ICESat era. There is a similar area that is also lowering on

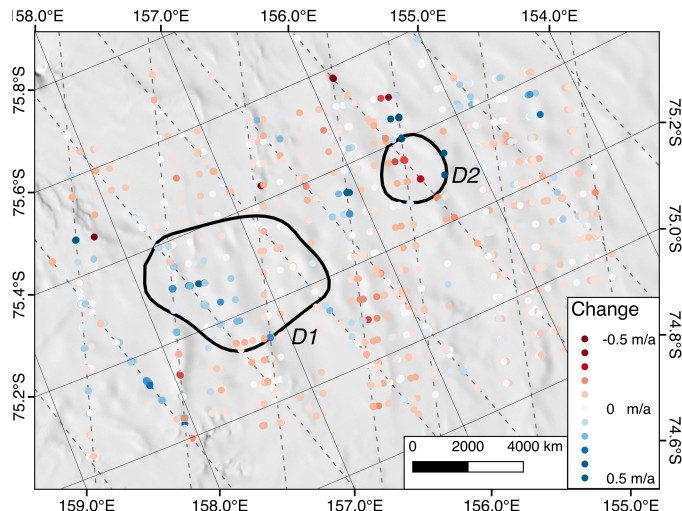

**Figure 6.** Difference between the 2017 KOPRI surface elevations and the 2004-2009 ICESat elevations, normalized by time between observations. Blue is rising, red is falling. Dashed lines are nominal ICESat ground tracks. Background image is the MODIS mosaic.

the south side of the large positive anomaly, and all three extrema are observed in both profile and intersection data. Note that the two points to the west of D2 where the surface appears to be rising are from a single not-repeated GLAS track, so there is no time series associated with them and we do not consider them to be a reliable signal.

Unfortunately, this survey alone is unable to address the $\frac{dH}{dt}$ behavior in detail since the end of the the ICESat era. However, there is no evidence that any of the Smith et al. (2009) lakes have switched from draining to filling or vice versa, and they have been established to be at a high stand relative to previous ICESat observations. Additionally, there is no evidence of ice-dynamic associated $\frac{dH}{dt}$ signal in the David Glacier region when compared to ICESat data. That is, patterns of surface elevation lowering are not associated with surface velocities or their gradients.

## 4.2 Hydraulic Potential Gradients

Figure 8 shows gridded hydraulic potential over the survey. The most immediate observation in the new hydraulic potential map is that there is a ridge running through D1. This is consistent with surface observations of crevassing extending into D1, and it confirms that lake outlines based on interpolating between repeat-track surface elevation changes do not necessarily correspond to a large connected collection of water, which by definition would have to be at a constant potential. However, there is a broad low over the lower part of D1, which is consistent with the re-interpreted surface elevation record. Additionally, there is no clear potential minimum associated with the original D2 outline. Instead, this survey shows a small minimum to the south.

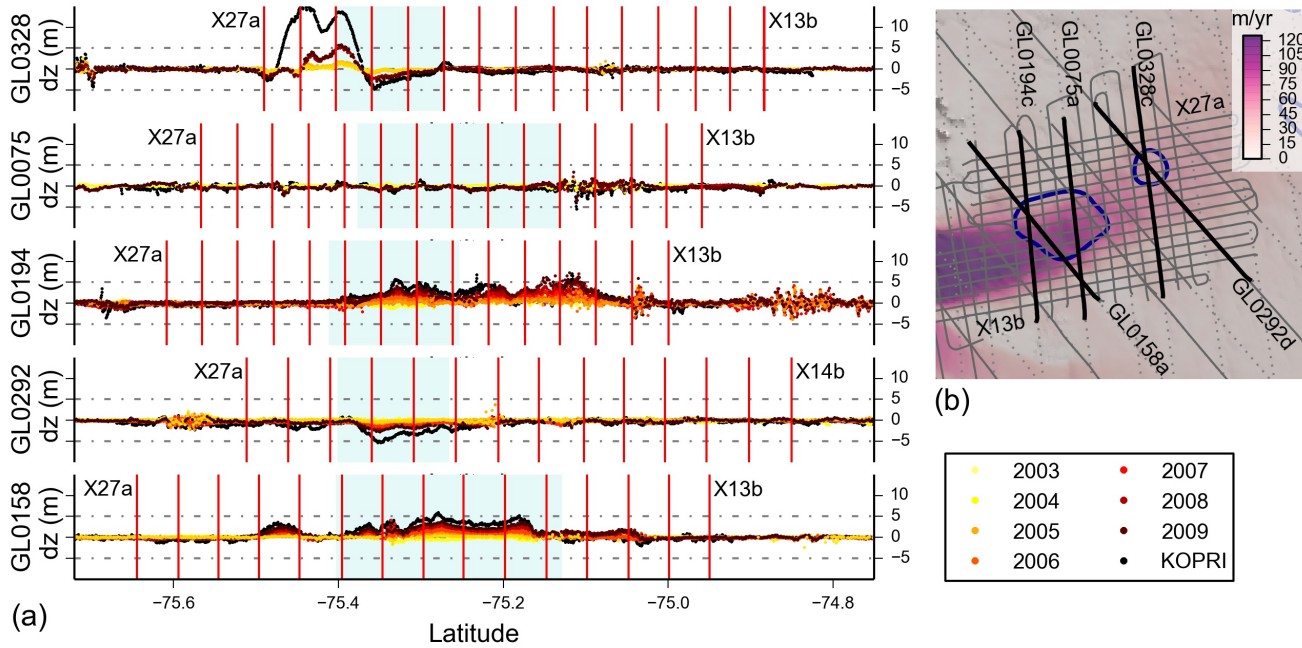

**Figure 7. (a)**:Individual profiles for the GLAS reflights showing surface elevations with respect to modeled surface in 2005. Vertical red lines show the intersections with the X transects, and blue shading shows where the profiles intersect the Smith et al. (2009) lakes. **(b)** Context map showing locations of selected tracks. Background is the ice surface velocity (Rignot et al., 2011b).

## 4.3 Reflection Coefficient

Figure 9 shows the reflection coefficients that have been corrected for geometry and ice loss. While the highest reflection coefficients are in the main trunk of the glacier and found in areas of greater than ∼1700 m ice thickness, their distribution within those bounds is not obviously correlated with surface velocities or ice thicknesses. Instead, around lake D1, $P_r$ tends to
5   be higher in regions with lower gradients of hydraulic potential, consistent with water pooling. The region around lake D2 is more complicated, with bright beds corresponding to low hydraulic potential gradients, but not necessarily matching up with the observed surface deflections.

It is possible that a more sophisticated method of calculating the contribution of dielectric ice loss would lead to clearer results. Model-based approaches were not pursued: the continent-wide model from Matsuoka et al. (2012) has insufficient
10   resolution, and integrating a dynamic model with the new topography is beyond the scope of this paper.

We also note that the span of reflection coefficients is still larger than would be expected for typical materials, and cannot be explained by contributions of dielectric ice loss alone. The analysis presented here used the radar equation for specular interfaces; a pure scattering interface would have a geometric spreading loss of $\frac{1}{r^4}$ (Peters et al., 2005). Additionally, there

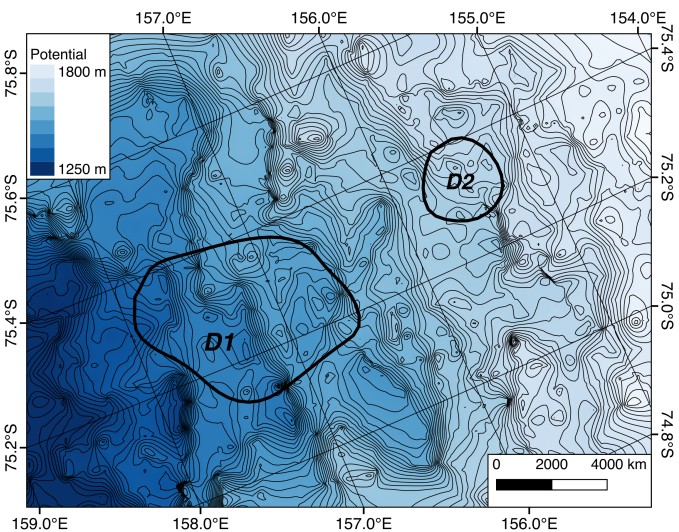

**Figure 8.** DEM showing hydraulic potential with 10 m contours for the region around lakes D1 and D2. The uncertainty is estimated as at least 10 m of hydraulic head, or one small contour line.

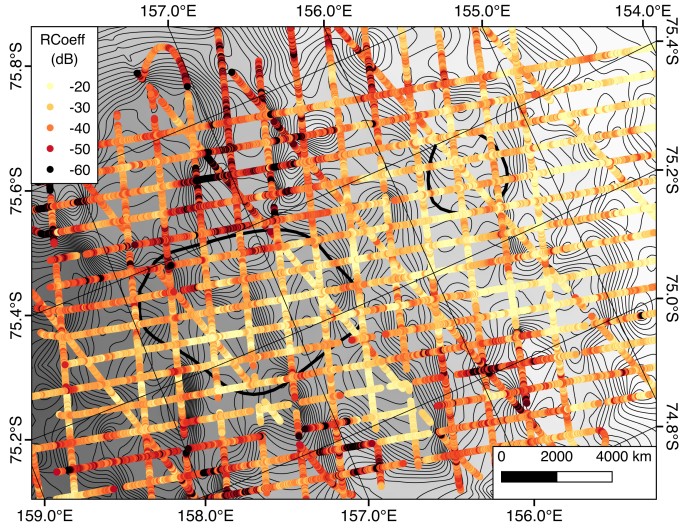

**Figure 9.** Relative reflection coefficients for the lake region, corrected for geometry and one-way dielectric losses of 15.1 $\frac{dB}{km}$. Background is 10 m contours for hydraulic potential, with hypothesized lake locations outlined in black (Smith et al., 2009).

could be englacial or surface terms not correlated with ice thickness that we are not accounting for. There is significant surface crevassing along the shear margins and over parts of D2, so correcting for surface scattering losses (Schroeder et al., 2016a) will likely yield an improvement in reflection coefficients. This topic warrants future investigation.

## 4.4 Specularity

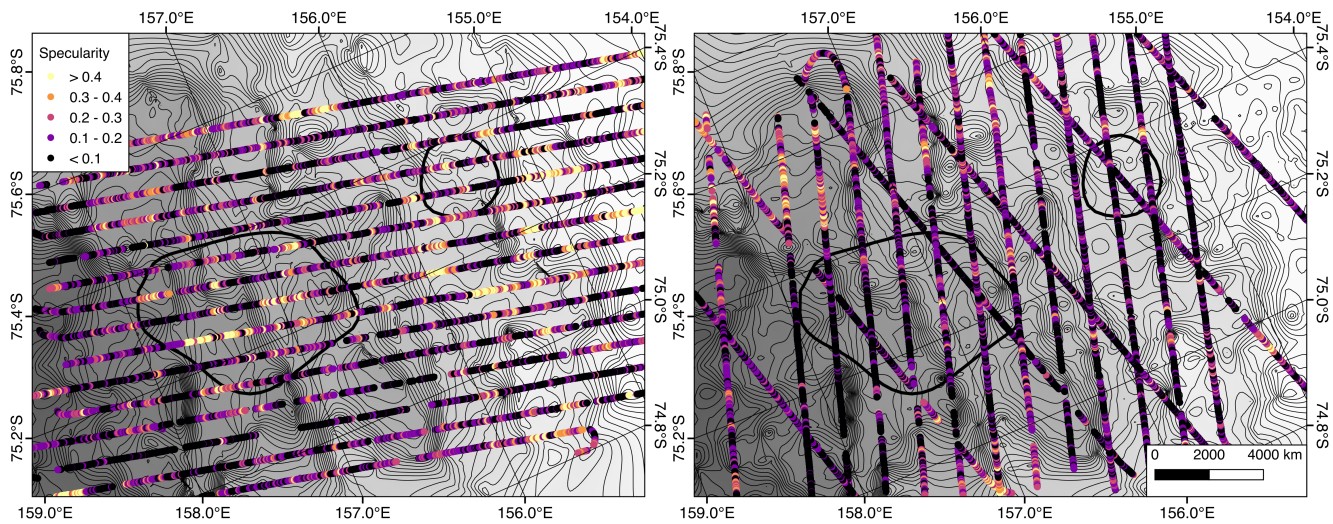

**Figure 10.** Specularity along **(a)** and across **(b)** flow of the main trunk. Background is the same as Fig. 9.

Figure 10 shows the results of calculating specularity content over the lake region, plotted on top of the hydraulic potential contours. As with the reflection coefficient analysis, the regions of higher along-flow specularity are aligned with the regions of lower gradient. This is consistent with water collecting where it is flat, then being transported more efficiently under higher gradients. However, the clear anisotropy in the specularity signal is not characteristic of a typical radar lake, which would be expected to have an isotropically mirror-like surface (Young et al., 2016). Instead, we see higher specularity along flow, and lower across flow.

## 5 Discussion

This paper presents results from a survey of lower David Glacier that includes surface elevation, subglacial topography, and radar-derived boundary conditions of potential active subglacial lakes. Beyond providing first-order boundary conditions for modeling and for site selection for a drilling campaign, this paper uses the new surface elevation changes and grids of hydraulic potential to suggest new locations for the lakes and provides an initial look at the radar-derived basal properties that differentiate active lakes from traditional radar lakes.

### 5.1 Reinterpretation of lake locations

Smith et al. (2009)'s classification of D1 was based on 3 lines and D2 was based on a single line. Their paper does not specify which GLAS orbits were used, but based on the quality of the data, D1's outline was presumably derived from GL0194,

GL0158, and GL0039 but not GL0075; and D2 was based on GL0292 and not GL0328. Lines GL0075 and GL0328 would have been left out of the Smith analysis due to having insufficient repeats for determination of cross-track slope.

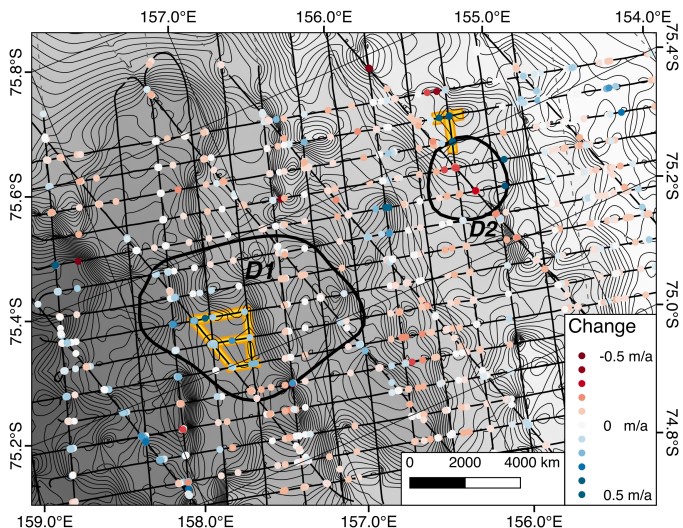

**Figure 11.** Observed time-normalized surface elevation changes are plotted on a DEM showing hydraulic potential with 10 m contours for the region around lakes D1 and D2. Transect segments where this study observed correspondence between a hydropotential minimum and a time history of surface elevation change are highlighted in yellow.

Based on the new ice thickness and hydraulic potential grids discussed in Sect. 3.3.2, an extension of the ICESat surface elevation record, and an understanding of the sparse data spacing involved in the original outlines (Smith et al., 2009), this
5    paper concludes that the original D1 outline is transected by an across-flow ridge, with only the downstream portion associated with a significant surface elevation change signal. Additionally, the potential lakes do not extend as far south as the original outline.

Figure 11 shows time-normalized surface elevation change and hydraulic potential. It can be seen that the thought-to-be draining D2 is instead on the edge of a previously unexamined region with significant surface uplift that is not consistent with
10    the effects of ice dynamics. The area of maximum uplift is consistent with a local minimum in hydraulic potential to the south of the previous outline, and is bracketed by areas of significant, but smaller, subsidence. We interpret these as all being part of the same feature, corresponding to water accumulation. The surface expression of basal changes is not straightforward to determine: Sergienko et al. (2007) modeled this for a draining lake, and found an evolving, non-monopole pattern.

The analysis in our paper is able to include GLAS data from lines with fewer repeats and pointed farther off the nominal
15    tracks than Smith et al. (2009) could because the crossover analysis agrees with the profile-based comparisons, and none of the elevation change signals were correlated with cross-track distance or aircraft roll. So, instead of observing the larger elevation change, Smith et al. (2009) identified an outline for D2 based on a line that only crossed the bordering subsiding region. This caused them to infer an offset boundary from what we observe, as well as classify it as draining instead of filling.

Our results do not show any evidence of the D1 or D2 features switching between filling and draining, but neither can it be ruled out by a single snapshot seven years after the end of the GLAS era. Siegfried and Fricker (2018) used CryoSat-2 in an effort to extend the surface altimetry record for a large subset of Smith et al. (2009)'s original lakes, including David Glacier. They concluded that "small-magnitude height-anomalies on these lakes were in-phase with small height-anomalies in the region outside the lake boundaries", while pointing out that CryoSat-2 data is challenging to interpret in this region due to surface roughness. Their analysis does not agree with our results, where we see clear evidence of concentrated surface elevation change: we attribute the difference to laser altimetry's higher precision making it the preferred tool for this region.

This reclassification of D1 and D2's potential boundaries provides an illustration of the pitfalls inherent in attempting to study active lakes based purely on the Smith et al. (2009) polygons, or to use them to assume basal boundary conditions of temperature or basal shear stress. As described in the original paper, lake outlines are interpolations, based on data that is increasingly sparse farther north. This is particularly relevant for planning and interpreting surveys consisting of a single radar transect over an active lake — a traverse planned directly across the middle of D1 could easily have resulted in a transect crossing the region with high hydraulic potential gradients and no evidence of collected basal water in any form, missing the smaller region that has low hydraulic potential gradients and anisotropic specularity.

## 5.2 Radar signature of active lakes

Consistent with most other radar investigations of suspected active lakes (Welch et al., 2009; Wright et al., 2014; Langley et al., 2011), the D1 and D2 surface features are not associated with the relatively bright and isotropically specular signature of a classic radar lake. There are three possible explanations for this mismatch: surveys are looking for water in the wrong place, at the wrong time, or using the wrong features in the radargram. Previous investigations consisting of a single line could be explained by the complicated transfer function from basal changes to surface expression or by the uncertainty in the lake outlines. Other surveys that did not include surface elevation measurements could be explained by hypothesizing that the active lake was at a low stand. However, the laser altimetry presented here shows that both D1 and D2 are at an even higher stand than during the ICESat era, so inconclusive results cannot be explained as being due to drained lakes, and the spatial extent and density of the survey make entirely missing the lakes unlikely. Thus, we conclude that active lakes cannot be expected to share the distinguishing physical features of radar lakes.

In interpreting reflection coefficients, there are a number of possibly-complicating factors. In an active lake system, it is likely that there are significant portions of the bed at the pressure melting point (water needs to be flowing into them), which would lead to lower contrast between the ice/water interface and the ice/bed interface. Depending on the depth of the lake, the roughness of the water/rock interface, and the salinity of its water, it is also possible that the radar return from the water/rock interface could interfere with that from the ice/water interface, lowering the observed reflection coefficient MacGregor et al. (2011). Christianson et al. (2016) investigated anomalously low reflection coefficients in a region just offshore of the Whillans Ice Stream grounding zone, and concluded that they were due at least in part to sediments entrained in the ice not yet having melted out. Similarly, we could consider active lakes to be at one end of a continuum where stable radar lakes are the other end, and they are primarily differentiated by their water residence times. The more rapidly evolving features may not have

existed for long enough to melt a smooth roof, so we could be observing the preserved imprint of the bed at low stands or basal roughness advected from upstream of the lake. Supporting this view, some active lakes (Adventure Subglacial Trench) do appear on classic lake inventories, but they are typically the ones farther upstream, with longer cycle times.

Specularity is an appealing complement to reflection coefficient analysis for characterizing the basal interface. A classic lake would be expected to have a smooth, flat ice/water interface and appears as an isotropically specular surface, which requires decimeter scale smoothness over hundreds of meters. This concept has been used in earlier work to characterize the distribution of subglacial water: Young et al. (2016) looked at the anisotropy of individual lines by comparing the specularity of the first return to the amount of scattering recorded afterward, while Schroeder et al. (2013) leveraged a gridded survey of Thwaites Glacier. Schroeder et al. (2013) reported a pattern of anisotropic specularity in Thwaites Glacier, and concluded that it indicates "canals" of subglacial water pooling, aligned with the ice flow. Since canals require a sedimentary subglacial interface, this is consistent both with Carter et al. (2017)'s hypothesis that active subglacial lakes drain through canals and with Smith et al. (2017)'s observations of active lakes in the region of Thwaites Glacier where Schroeder et al. (2013) identified the water system transition. In the David Lakes region, we see an overall pattern of anisotropic specularity, including over the regions of surface elevation change, that is similar to that seen in Thwaites. Further work is needed to understand the overall hydrologic systems driving these active lakes, and the anisotropic specularity in this region provides an interesting constraint on possible organizations of water.

## 6   Conclusions

This paper describes a new aerogeophysical dataset focusing on the two most downstream active subglacial lakes on David Glacier. The primary sensors were a laser altimeter and ice penetrating radar. In combination, these collected data allow a determination of the surface elevation changes relative to the ICESat era, a higher resolution map of subglacial bed elevation, and the first radar-derived characterization of this region's ice/bed interface.

First, comparing new laser altimeter surface elevations to the ICESat record shows that the original Smith et al. (2009) lake outlines require refinement (Fig. 6). The most downstream lake (D1) has continued to fill, but its extent is probably smaller than the original outline. The second-most downstream lake (D2) was originally classified as draining. However, the new surface elevation data reveals a larger anomaly adjacent to the original D2 outline. This anomaly appears to be a filling lake, and D2 to be an edge effect.

Next, ice penetrating radar data was used to estimate the basal hydraulic potential in the David Lakes region. Lake D1 is divided by a clear hydraulic potential ridge, with the downstream portion corresponding to the largest area of surface elevation change. The upstream part of D1 has a lower amplitude surface elevation signal, primarily appearing in the profile data. Additionally, there are nearby areas of hydraulic potential minima that do not appear to have a surface elevation signal. The story around D2 is less clear, but the highest amplitude surface elevation changes appear to be associated with a shallow hydraulic potential minimum.

Traditionally, the basal reflection coefficient has been a primary tool in identifying subglacial water. This paper attempted to sidestep the well-known pitfalls inherent in calculating absolute basal reflection coefficients and instead focused on selecting a dielectric ice loss that would lead to acceptable uncertainties in the relative reflection coefficients. Consistent with previous radar surveys of active lakes, neither D1 nor D2 would be categorized as a classic radar lake on the basis of relative reflection coefficients. There is a weak correspondence between regions of low hydraulic potential gradient and elevated basal reflection coefficients, but the association is inconclusive and the results neither confirm nor rule out the existence of concentrated subglacial water. In the case of active lakes, this would make sense if they are part of a distributed water system on a wet bed.

Finally, we looked at the specularity content of the basal interface. Rather than being isotropically specular, as would be expected for an extensive subglacial lake, it is anisotropically specular, with high specularity occurring along-flow. Both the specularity and reflection coefficient signals are strongest near the lower portion of lake D1, while the region around D2 is more ambiguous with high reflection coefficients and anisotropic specularity distributed across the glacier's trunk. The anisotropic specularity seen here is similar to observations on Thwaites Glacier in the region of its newly-discovered active lakes. This radar signature could be consistent with either water accumulating in linearly organized features or with the active lakes' roofs retaining the imprint of the deflated state even as they are filling.

*Data availability.* Profile based survey data of ice thickness, surface elevation, bed elevation, radargrams, and positioning data will all be available in Zenodo upon publication. (DOIs to be provided before publication.)

*Author contributions.* LEL: Formal analysis, writing, visualization, methodology, conceptualization. LHB: writing, visualization, conceptualization. DAY: data curation, formal analysis, methodology, writing. ESQ: investigation, writing. DDB: funding acquisition, conceptualization, writing. CKL: funding acquisition, conceptualization, writing. WSL: funding acquisition, conceptualization, writing. JIL: funding acquisition, conceptualization, writing. JHL: funding acquisition, conceptualization, writing.

*Competing interests.* The authors declare that they have no conflict of interest.

*Acknowledgements.* We acknowledge the support of the G. Unger Vetlesen Foundation. LEL was partially supported by the UTIG Ewing-Worzel Fellowship and the Gale White Endowed Fellowship in Geophysics. WSL and CKL were supported by a research grant from the Korean Ministry of Oceans and Fisheries (KIMST20190361; PM19020). JIL was supported by a research grant from Korea Polar Research Institute (PE20020). JHL was supported by a research grant from the Korea Polar Research Institute (PE20050). The authors would like to thank Helicopters New Zealand pilot Phil Robinson and engineer Fred Wunderler for their support and dedication to this project. Dillion Buhl, Tom Richter, Greg Ng, and Scott Kempf provided invaluable engineering, technical, and field support. We thank Joseph MacGregor and Nicholas Holschuh for their thoughtful and constructive reviews. This is UTIG contribution 3647.

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
