# Peer review of "Aerogeophysical characterization of an active subglacial lake system in the David Glacier catchment, Antarctica"

_The Cryosphere, 2019_

## Referee Comment (RC1) · Nicholas Holschuh (Referee) · 16 Nov 2019

**Review of:** *Aerogeophysical characterization of an active subglacial lake system in the David Glacier catchment, Antarctica*
**Submitted to:** The Cryosphere
**Reviewer:** Nicholas Holschuh

**General Comments:**
This paper provides a review of data collected over David Glacier from a helicopter variant of the UTIG HiCARS radar system. The authors provide an honest interpretation of the data, acknowledging the limits of radar's ability to diagnose subglacial hydrologic characteristics, and add to the growing body of literature highlighting discrepancies between altimetry and radar delineated subglacial lakes. The work is well written and thorough. There is room to expand on the interpretations in a few key areas (mentioned below), but overall I found this to be a nice contribution to the literature and nearly ready for publication.

**Technical Comments:**
I appreciated the thorough discussion of attenuation in this work, but a slight reorganization can make the caveats of the applied method clearer. On page 12, line 25, you state that you assume that basal reflection coefficients are independent of ice thickness, but then spend much of the later sections articulating why that assumption is wrong, which requires you to restrict the data used in your attenuation fit. I think it would be more transparent to discuss the basis for assuming that both depth-averaged attenuation rate and basal reflectivity might be depth-correlated (presently stated at P.13, L3-5, and P.13, L15-16), and then proceed with the method in the face of those known caveats. This would provide a succinct description for future users of the method, and make it clearer to the reader why your final estimate is a lower bound.

It is worth mentioning, based on the results presented in figure 9, that your final returned-power distribution spans 70 dB. This represents a major challenge in radar interpretation more broadly, and one I have faced in my own data interpretation. As you point out, the range of reasonable reflectivity variation due to variation in dielectric properties of the substrate material is ~20-30 dB, not 70 dB. The fact that variance is so high means we are missing a critical (and potentially, dominant) control on the values. It might be worth pointing this out to the reader.

When discussing why altimetry derived lakes may not be obvious in radar data, I think it is important to keep in mind that both radar system characteristics and specific lake characteristics (such as water depth) will affect how the lakes present in radar data. Different radar systems will have different sensitivities to what is likely a double interface (ice-water, water-rock), with water column thickness either (a) falling under the range resolution of the system over the whole lake, or (b) pinching out to a water-layer thickness below the range resolution at the lake margins. You know from the altimetry data presented in figure 4 that the water layer thickness changes on the order of meters (not tens of meters), which means that thin film effects may provide an (at least partial) explanation the character of lakes in both this survey and those in the literature (see Christianson et al., 2016 for more details on thin film effects in radar). You would expect lakes across surveys to appear different, given the range the system characteristics for radars used in the papers cited:

- Impulse 3 MHz system: 28 m range resolution in ice, 8 m in water (Welch et al., 2009) (Langley et al., 2011)

- UTIG/SOAR data, HiCARS 4MHz Bandwidth: 21 m range resolution in ice, 6.25 m in water (Carter et al., 2009)(Langley et al., 2011)
- HiCARS 14MHz Bandwidth: 6 m range resolution in ice, 1.8 m in water) (Wright et al., 2014)

A discussion of a damped signal from thin lakes or the effect of water depth more generally would be a useful addition somewhere in this paper. This feeds back into roughness interpretation, as a thin lake lid may look quite rough (as mentioned in the very last line of the conclusions).

Finally, I think the discussion of lake D2 should be made clearer. In several places, you state that the behavior hasn't changed since the Smith et al. paper (e.g., P.14, L.27) but it is not clear how to interpret that statement given that part of the surface in that region is lowering and part of it is rising (rapidly). Perhaps there is a geolocation error in Smith et al., 2019, or the lake boundaries have changed since then? In general, I think your observations around D2 are fascinating, and warrant further discussion.

**Line-Item Corrections:**

| | |
|---|---|
| Page #: 1-4
Line #: | I think you've done a really nice job of summarizing the literature here - this is a great resource to provide the community. |
| Page #: 9
Line #: 7-8 | It would be helpful if you put numbers on the uncertainty evaluation here (along-track bed elevation variation [height over width] and cross-track beam width) |
| Page #: 8
Line #: 1 (fig 4) | Is the dz presented here normalized by the dt between observations? If not, it should be, so that you can reasonable compare observations that span 2003-2017 and those that span 2009-2017. |
| Page #: 8
Line #: 13 | The phrasing of this sentence is a bit awkward - perhaps: "Due to the existence of side lobes in the transmit/receive beam the first return criteria may understeimate ice thickness in rough terrain." |
| Page #: 9
Line #: 5-8 | Additional context would help the reader understand these errors. Define what it means to be "high bed slope" |
| Page #: 11
Line #: 1-3 | It would be useful to show how these errors might affect hydrualic potential near the lake boundaries. One way to do that would be to present an error map. |
| Page #: 11
Line #: 7 | The clause "cyclical active lakes are manifestations of feedback loops within this system" feels out of place, I suggest removing it. |

| | |
|---|---|
| Page #: 11
Line #: 24-26 | 10m of hydraulic head seems like a very low uncertainty given the bed-error estimates presented above. More justification is required before you quote that number, especially given the following sentence. |
| Page #: 12
Line #: 16-24 | In this case, you have a known calibration surface in those data that span Drygalski Ice Tongue shown in figure 1. Using that surface would allow you to bolster your reflectivity analysis - if you are choosing not to use it in this study, you should justify that choice here. |
| Page #: 12
Line #: 25-28 | Here is where it would be useful to introduce the known sources of errors in this method (instead of starting with the assumption that they don't exist). As you state, temperature is correlated with ice thickness, and water content is correlated with temperature, introducing two factors that will complicate your interpretation of attenuation rate using the data alone. |
| Page #: 8
Line #: 1 | Figures 4-8 might make more sense in the results section than the methods section. |
| Page #: 13
Line #: 15-16 | It might be useful to do a back of the envelope calculation to show that 1300 m is deep enough to expect a melted bed here. |
| Page #: 14
Line #: 8 | Here, in your discussion of sources of variance in reflection coefficient, you should introduce the idea of thin films / the effect of water depth. |
| Page #: 14
Line #: 26-27 | Here is an opportunity to clarify your interpretation of lake D2 dynamics. The statement that it hasn't switched from draining implies you think the surface elevation gain on the edge of the formerly defined lake (which corresponds with the hydropotential basin) is not a change in behavior. More text here to help the reader understand your position would be useful. |
| Page #: 15
Line #: 14-17 | I find this statement confusing, and I think it is because I don't know what "the region of interest" is. Do you mean the lakes? The places where ice thicknesses are the highest don't seem to be over the lakes, but are where you would expect bed power to be the highest (as they fall well above the solid lie in Figure 8). It would help to clarify this point. |
| Page #: 17
Line #: 6-8 | Do you provide a new outline for lake D2? That would be a useful outcome of this study. Also, this part of the text is an opportunity to clarify your thinking, is it draining or is it filling? Is this different from its behavior as stated by Smith et al.? |

Page #: 17
Line #: 16-17

Again, do you think D2 is still draining? Despite the huge positive anomaly nearby? More explanation is required.

Page #: 18
Line #: 8-9

This statement is hard to defend -- can you imagine lifting the ice surface up 10 m through changes in water storage but not fully decoupling the ice from the rock to form a lake? You seem to argue this is analogous to what is observed at Thwaites, but I find that section of the text difficult to follow. If you want to keep this interpretation in, more discussion / justification is required.

Page #: 19
Line #: 21-22

This point is a really interesting one, and I think should come up in the discussion earlier so the reader is prepared for it in the conclusions. Perhaps introduce this idea up near P.14, L,8.

---

## Referee Comment (RC2) · Joseph MacGregor (Referee) · 8 Dec 2019

Joseph A. MacGregor 9 December 2019

Summary

This manuscript describes a new aerogeophysical survey of the area upstream of the David Glacier terminus near the South Korean base there. It describes in detail the scientific motivations, instruments and platform used, surveys, data and implications. A key result is not necessarily a new one – that subglacial lakes appear quite different in altimetry vs. radar sounding. However, the lake system considered includes two goodsize, candidate lakes, the survey is high-resolution, and the conclusions are perhaps stronger than is possible from previous comparable studies.

As a whole, this manuscript is quite impressive. It is not easy to relatively concisely convey as much meaningful information as done in this manuscript. From a bird's eye view, the manuscript summarizes an entire, substantial aerogeophysical survey of an area that has heretofore received less attention, carefully informs the reader and outlines a sincere attempt at reconciling the mismatch between altimetry radar sounding over active subglacial lakes. Overall, it represents a substantial contribution and is appropriate to The Cryosphere. In the details, the manuscript rarely misses a beat. I have only a few concerns listed below and cannot consider any of them more than minor.

Comments

2/5: Explain more clearly here why Siegfried and Fricker (2018) didn't consider these to be "true" lake features. This is reconsidered later on but is awkward as left hanging here.

6/10-16: Given that a new antenna system is introduced, a slightly more detailed diagram illustrating the antenna configuration might be appropriate. In particular, I'm wondering whether the forward-pointing boom is empty?

8/9: Is this the "2-D focusing" described in the earlier Peters et al. studies? Avoid sending the reader all the way to the reference describing this key point of the processing sequence.

9/11: Why not use ordinary kriging? The data appear extensive enough.

Section 3.3.3 Reflection Coefficients: A thoughtful discussion of attenuation issues and in particular the roll-over shown in Figure 8. I find the sub-division of this power–thickness relation plausible. A minor concern is that the uncertainty on the best-fit attenuation rate is not reported, which could be used to further evaluate the significance

of the spatial variation in bed reflectivity. It's probably not a significant effect upon the spatial variation, but it should still be reported.

20/23: This completely fair statement then begs the question as to why that calibration wasn't done. As I recall, there is often open water within helicopter range of the basing during the Antarctic summertime. Please clarify.

14/5: Compare to regional value reported by Matsuoka et al. (2012). I respect the later argument that the Matsuoka et al. (2012) grid is coarse compared to this survey's grid, but a nominal comparison should still be possible.

15/7: Because a graticule is consistently used in the figures (which I like), revise the phrasing "grid north" to (probably) "south". A north arrow in the figures could ameliorate the situation.

Discussion is excellent.

Figures

Figures 1a/6/7: Add scale bar. Figure 6/7: Label lakes as D1/2.

Figure 3: The radargram contrast is quite poor on paper and in print. Color range as mean +/- two standard deviations usually works well. Is the very top of the radargram indeed the surface, following elevation correction?

Figure 8: Narrow vertical scale (remove lower 20 dB).

Figure 9: Narrow the color scale used significantly. It's ok to saturate if less than / greater than symbols are used at the edge of the range. Otherwise, I'm not getting much out this reflectivity map.

Grammar, etc.

1/7: While understandable if perhaps unintentional, it isn't necessary to be quite so dismissive of one's own (significant) efforts in an abstract. I suggest changing the

sentence that starts with "While. . ." to more positively reflect the effects undertaken and drop the "not the first" statement.

Introduction as a whole is perhaps a quarter too long and more appropriate to a dissertation chapter than a journal article. Review to simplify further.

Throughout: CryoSat-2 not CryoSat.

2/7 to 3/4: Merge these two paragraphs.

9/3: "policy" is an awkward choice of terms here. Perhaps clarify what first-return picking means relative to other options?

11/15-17: In the phrasing used here, "X%" makes more sense to describe the fraction of overburden pressure that earlier studies considered.

12/10: Cite Matsuoka et al. (2012)

12/14: "Some studies attempt. . ." (some of these papers include some of the same authors even though they used different methods)

12/17: "David Glacier region"

Figure 8 caption: "we assume" not "we think"

13/11-10 and 13/15: Description of graphical elements in figures (as opposed to what the displayed data mean for interpretation) should be reserved exclusively to figure captions.

17/17: "a single snapshot seven years"

---

## Author Comment (AC1) · 30 Apr 2020

**Author's Response**

We thank the reviewers for their thorough, constructive, and encouraging feedback.

Responses to individual comments are in the attached document.

**Abstract/Introduction**

*R2: 1/7: While understandable if perhaps unintentional, it isn't necessary to be quite so dismissive of one's own (significant) efforts in an abstract. I suggest changing the sentence that starts with "While..." to more positively reflect the effects undertaken and drop the "not the first" statement.*

Thank you. We have re-worded the abstract as suggested.

*R1: p1-4/*: I think you've done a really nice job of summarizing the literature here - this is a great resource to provide the community.*

*R2: Introduction as a whole is perhaps a quarter too long and more appropriate to a dissertation chapter than a journal article. Review to simplify further.*

Given the differing feedback from the reviewers regarding the introduction, we have chosen not to substantially edit it.

*R2: 2/5: Explain more clearly here why Siegfried and Fricker (2018) didn't consider these to be "true" lake features. This is reconsidered later on but is awkward as left hanging here.*

We have changed the wording to clarify the criteria Siegfried & Fricker (2018) used in identifying "true" lakes. New text:

*A more recent analysis using CryoSat-2 data compared the patterns of surface elevation change within and outside the D1 and D2 lake polygons and concluded that these might not be true lake features because the changes were small and did not have a phase difference across the nominal lake boundaries \citep{Siegfried2018}.*

*R2: 6/10-16: Given that a new antenna system is introduced, a slightly more detailed diagram illustrating the antenna configuration might be appropriate. In particular, I'm wondering whether the forward-pointing boom is empty?*

We had originally mentioned the empty forward boom in the caption for Figure 2; we now address that here as well. The relevant sentences now read:

*The helicopter's antennas were designed to fit inside existing flight-certified booms originally designed for magnetometer surveys. These geometric constraints led to an end-fed design with an end plate installed in each lateral boom; the forward boom was empty.*

**Overall Organization:**

*R1: p.8/1: Figures 4-8 might make more sense in the results section than the methods section.*

***Editor: The data and their analysis are well-presented, the methods also. Maybe a clearer separation between the methods and results section could be obtained, for example by moving some of the figures in the results section.***

We have moved figures 4 & 5 (now 6 & 7) into section 4.1 (Results: Lake Stage) and Figure 7 (now 8) into Section 4.2 (Results: Hydraulic Potential Gradients).

Figures 6 & 8 (now 4 & 5) help to motivate some of the discussion within methods, so were left in Section 3.

**Methods: Radar Processing**

***8/9: Is this the "2-D focusing" described in the earlier Peters et al. studies? Avoid sending the reader all the way to the reference describing this key point of the processing sequence.***

We added a description of focusing to the text:

*For this work, we used the 1D-focused processing for radargrams described in Peters et al. (2007a) for geometry and basal reflectivity, complemented with 2D focusing to derive specularity content (Schroeder et al., 2015). Figure 3 shows an example radargram that crosses D2. Focusing is performed by convolving a kernel with pulse-compressed radar data, where the kernel is generated based on the expected appearance (delay and phase) of a point scatterer at that location, which is a function of airplane height, ice thickness and surface slope. Different aperture lengths are used for focusing, which correspond to the 1D and 2D nomenclature in Peters et al. (2007a). 1D focusing uses a short enough aperture that range changes are less than a pulse pulse width; for a longer aperture, a 2D kernel (in this case accommodating 1 $\mu$sec of range change) is required to match the phase history, further improving resolution, collection of scattered energy and detection of sloping interfaces with some cost to signal to noise ratio.*

**Methods: Bed elevation + hydraulic potential**

***R1: p9/7-8: It would be helpful if you put numbers on the uncertainty evaluation here (along-track bed elevation variation [height over width] and cross-track beam width)***

We provide numbers for total uncertainty.

In order to address R1's request to be able to interpret how the distribution of errors would affect lake boundary interpretation, we now plot the observed crossover errors on the gridded bed elevations.

[Figure]

**R1: p9/5-8: Additional context would help the reader understand these errors. Define what it means to be "high bed slope"**

Rewrote this sentence to clarify.

**R2: 9/11: Why not use ordinary kriging? The data appear extensive enough.**

We tried a number of gridding methods with the goal of reducing artifacts due to the high density of data along-track compared to across-track. We are providing the raw profile based data so other researchers can generate grids that are appropriate for their uses.

**R1: p11/1-3: It would be useful to show how these errors might affect hydraulic potential near the lake boundaries. One way to do that would be to present an error map.**

We find the crossover errors to be more informative than the gridding errors for this type of analysis and have added them to the bed elevation map. Adding errors to the hydropotential map was too cluttered. The gridding errors follow the expected behavior of flattening slopes, but the maxima and minima locations do not change.

**Editor: 11.22 errors from satellite DEMs. Is it a result from this study? Or from earlier studies (then refer them).**

Satellite-derived DEMs were the wrong thing to be comparing to here; should have been radar surface elevations. We have changed this.

**R1: p11/24-26: 10m of hydraulic head seems like a very low uncertainty given the bed-error estimates presented above. More justification is required before you quote that number, especially given the following sentence.**

Given that it does seem counterintuitively low, we have added more details to the text describing how it was derived.

(Note that the contour lines in Figure 6 for bed elevation are spaced at 100m, but those in figure 7 for hydraulic potential are spaced at 10m)

**Methods: Reflection Coefficients**

**R1: I appreciated the thorough discussion of attenuation in this work, but a slight reorganization can make the caveats of the applied method clearer. On page 12, line 25, you state that you assume that basal reflection coefficients are independent of ice thickness, but then spend much of the later sections articulating why that assumption is wrong, which requires you to restrict the data used in your attenuation fit. I think it would be more transparent to discuss the basis for assuming that both depth-averaged attenuation rate and basal reflectivity might be depth-correlated (presently stated at P.13, L3-5, and P.13, L15-16), and then proceed with the method in the face of those known caveats. This would provide a succinct description for future users of the method, and make it clearer to the reader why your final estimate is a lower bound.**

**R1: p.12/25-28:Here is where it would be useful to introduce the known sources of errors in this method (instead of starting with the assumption that they don't exist). As you state, temperature is correlated with ice thickness, and water content is correlated with temperature, introducing two factors that will complicate your interpretation of attenuation rate using the data alone.**

Thank you for pointing this out! We have reworded this paragraph, and think that the whole section works much better now. It now reads:

*Given these goals, we modified the simplest approach of determining depth-averaged dielectric ice loss from the slope of geometry-corrected echo strengths vs. ice thicknesses. This approach relies on the assumption that basal reflection coefficients are independent of ice thickness, which is overly simplifying since basal temperature, and therefore the presence of water at the bed, is correlated with ice thickness. We see evidence of a slope change associated with the likely presence of water, so restricted our linear regression to data in thinner ice. Since thinner ice is on average cooler than thicker ice, restricting the range of thicknesses used in the fit will result in an estimate that is a lower bound on dielectric ice loss.*

**R2 12/23: This completely fair statement then begs the question as to why that calibration wasn't done. As I recall, there is often open water within helicopter range of the basing during the Antarctic summertime. Please clarify.**

**R1: p.12/16-24: In this case, you have a known calibration surface in those data that span Drygalski Ice Tongue shown in figure 1. Using that surface would allow you to bolster your reflectivity analysis - if you are choosing not to use it in this study, you should justify that choice here.**

We intentionally left out details of an absolute calibration for the radar system, since our analysis only looks at relative reflection coefficients. (This is driven by uncertainties in ice loss and scattering.)

The Drygalski Ice Tongue is not a particularly good calibration target for several reasons: significant surface scattering from crevasses, hypothesized marine ice accretion, and probable differences in the temperature profile.

However, the data set does support several methods of calibration:

- As R2 points out, the best calibration for radar reflection coefficient is open water. We performed a test flight in New Zealand collecting data sufficiently far from shore. In Antarctica, HNZ has a policy against flying more than 500m over open water, making this more challenging. We did collect some data off the side of Nansen, but it has not yet been examined for suitability.
- We re-flew some transects that were previously flown with the HiCARS2 radar mounted on a DC3. There was an 11dB offset in basal reflection coefficients between the two systems, of which ~9dB can be explained by receiver differences (from lab-bench testing) and the presence of wings (ground plane effectively doubling transmit power). The HiCARS system is also routinely calibrated over water.

**R2: Section 3.3.3 A minor concern is that the uncertainty on the best-fit attenuation rate is not reported, which could be used to further evaluate the significance of the spatial variation in bed reflectivity.**

Added standard deviation to best-fit dB/km value.

**R1: p.13/15-16: It might be useful to do a back of the envelope calculation to show that 1300 m is deep enough to expect a melted bed here.**

We performed the requested calculation and updated the relevant figure. New text was added:

*Ice thickness required to reach the basal melting point can be estimated using the Robin model \citep{Robin1955, Cuffey2010}, which is a 1D model that accounts for ice thickness, accumulation rate, surface temperature, geothermal flux, and basal heat generation. There are many degrees of freedom and in an attempt to simplify and constrain the possible range of solutions a monte carlo approach was adopted.The accumulation rate, in ice equivalent, was assumed to have a normal distribution with one standard deviation of .06~$\pm$~.02 m/a \citep{VanWessem2014b}. Geothermal flux was assumed to have a normal distribution of .06~$\pm$~.01~$\frac{W}{m^2}$ \citep{An2015}. Surface temperature is assumed to have a normal distribution of $-35 \pm 5 ^\circC$\citep{VanWessem2014a}.Frictional heating was added as a constant to geothermal flux with magnitudes of 0, 0.01, or 0.03~$\frac{W}{m^2}$ for three separate model runs. These value of frictional heat is appropriate for basal sliding of 50 to 100~m/a \citep{Rignot2017} and 10 to 20~kPa of basal shear stress. Twenty thousand solutions*

*were generated, and the resulting cumulative probabilities of a thawed bed using the moderate frictional heating magnitude are shown in Figure~\ref{fig:ice_loss_calculation}.*

**R2: 14/5: Compare [dielectric ice loss] to regional value reported by Matsuoka et al. (2012). I respect the later argument that the Matsuoka et al. (2012) grid is coarse compared to this survey's grid, but a nominal comparison should still be possible.**

Unfortunately, the data from Matsuoka 2012 does not appear to be publicly available. Eyeballing the maps gives <= 5dB/km for the pure-ice contribution, but the acid contribution is expected to be significant in this region (maybe even equal to the pure ice, but no number is given). This comparison to Matsuoka2012 seems too weak to include in a meaningful way in this paper.

**R1: It is worth mentioning, based on the results presented in figure 9, that your final returned-power distribution spans 70 dB. This represents a major challenge in radar interpretation more broadly, and one I have faced in my own data interpretation. As you point out, the range of reasonable reflectivity variation due to variation in dielectric properties of the substrate material is ~20-30 dB, not 70 dB. The fact that variance is so high means we are missing a critical (and potentially, dominant) control on the values. It might be worth pointing this out to the reader.**

We have more explicitly highlighted this observation in the Results section. Relevant paragraph now reads:

*We also note that the span of reflection coefficients is still larger than would be expected for typical materials, and cannot be explained by contributions of dielectric ice loss alone. The analysis presented here used the radar equation for specular interfaces; a pure scattering interface would have a geometric spreading loss of $1/r^4$ (Peters et al. 2005). Additionally, there could be englacial or surface terms not correlated with ice thickness that we are not accounting for. There is significant surface crevassing along the shear margins and over parts of D2, so correcting for surface scattering losses (Schroeder et al., 2016a) will likely yield an improvement in reflection coefficients. This topic warrants future investigation.*

**R1: p.15/14-17: I find this statement confusing, and I think it is because I don't know what "the region of interest" is. Do you mean the lakes? The places where ice thicknesses are the highest don't seem to be over the lakes, but are where you would expect bed power to be the highest (as they fall well above the solid line in Figure 8). It would help to clarify this point.**

We agree that the statement was confusing and don't think it added anything to the analysis.

Deleted.

**Results & Discussion**

*Editor: Section 5.1 Showing an outline of the refined lake locations compared to the Smith et al. outlines would be welcome.*

*R1: p.17/6-8:Do you provide a new outline for lake D2? That would be a useful outcome of this study.*

We have added a figure that shows surface elevation changes in context with the gridded hydraulic potential, and highlighted segments of our survey lines where a time series of surface expression change corresponds to a hydraulic minimum:

[Figure]

*R1: Finally, I think the discussion of lake D2 should be made clearer. In several places, you state that the behavior hasn't changed since the Smith et al. paper (e.g., P.14, L.27) but it is not clear how to interpret that statement given that part of the surface in that region is lowering and part of it is rising (rapidly). Perhaps there is a geolocation error in Smith et al., 2019, or the lake boundaries have changed since then? In general, I think your observations around D2 are fascinating, and warrant further discussion.*

We think that the observed lowering is an edge effect around a rapidly rising center.

Smith et al. 2009 does not explicitly address which individual GLAS lines were used in their analysis. (Assuming 2019 was a typo?)

GL0328 was the only GLAS line that the rising area south of D2 appeared in. However, based on the Smith2009 criteria for including data, we suspect that GL0328 had insufficient along-track repeats to be included in their analysis, and that D2 was based solely on GL0292. Since we are able to validate our along-track dz/dt numbers with the crossing lines, we do not rely as heavily on repeats for surface slope corrections. We were able to use that line because the crossing

lines in our survey validated that the apparent elevation change was real and not due to tracking error + surface slope.

So, instead of finding the larger elevation change, Smith09 identified an outline for D2 based on a line that only crossed the boundary effect. This caused them to have an offset boundary for the "lake", as well as classify it as draining instead of filling, as we observe.

This was apparently a point of some confusion; we have addressed it in the manuscript by adding a figure and by updating the text as described in the responses to the next 3 comments.

***R1: p.14/26-27: Here is an opportunity to clarify your interpretation of lake D2 dynamics. The statement that it hasn't switched from draining implies you think the surface elevation gain on the edge of the formerly defined lake (which corresponds with the hydropotential basin) is not a change in behavior. More text here to help the reader understand your position would be useful.***

In the Results section, we have re-written this paragraph to call attention to the two lowering regions that bracket a larger raising region, attempting to better set up our conclusion in the Discussion section. The new text describing these features reads:

*While the region inside the original D2 outline appears to still be lowering with a total displacement of $\sim$5~m, it borders an area along line GL0328 with up to 15~m of elevation \textit{gain} since the ICESat era. There is a similar area that is also lowering on the south side of the large positive anomaly, and all three extrema are observed in both profile and intersection data. Note that the two points to the west of D2 where the surface appears to be rising are from a single not-repeated GLAS track, so there is no time series associated with them and we do not consider them to be a reliable signal.*

***R1: p.17/6-8: Also, this part of the text is an opportunity to clarify your thinking, is it draining or is it filling? Is this different from its behavior as stated by Smith et al.?***

***R1: P.17/16-17:Again, do you think D2 is still draining? Despite the huge positive anomaly nearby? More explanation is required.***

Attempted to clarify the text. It now reads:

*The thought-to-be draining D2 is instead on the edge of a previously unexamined region with significant surface uplift that is not consistent with the effects of ice dynamics. The area of maximum uplift is consistent with a local minimum in hydraulic potential to the south of the previous outline, and is bracketed by areas of significant, but smaller, subsidence. We interpret these as all being part of the same feature, corresponding to water accumulation. The surface expression of basal changes is not straightforward to determine: \cite{Sergienko2007} modeled this for a draining lake, and found an evolving, non-monopole pattern.*

*The analysis in our paper is able to include GLAS data from lines with fewer repeats and pointed farther off the nominal tracks than \cite{Smith2009} could because the crossover analysis agrees with the profile-based comparisons, and none of the elevation change signals*

*were correlated with cross-track distance or aircraft roll.So, instead of observing the larger elevation change, \cite{Smith2009} identified an outline for D2 based on a line that only crossed the bordering subsiding region. This caused them to infer an offset boundary from what we observe, as well as classify the feature as draining instead of filling.*

*When discussing why altimetry derived lakes may not be obvious in radar data, I think it is important to keep in mind that both radar system characteristics and specific lake characteristics (such as water depth) will affect how the lakes present in radar data. Different radar systems will have different sensitivities to what is likely a double interface (ice-water, water-rock), with water column thickness either (a) falling under the range resolution of the system over the whole lake, or (b) pinching out to a water-layer thickness below the range resolution at the lake margins. You know from the altimetry data presented in figure 4 that the water layer thickness changes on the order of meters (not tens of meters), which means that thin film effects may provide an (at least partial) explanation the character of lakes in both this survey and those in the literature (see Christianson et al., 2016 for more details on thin film effects in radar). You would expect lakes across surveys to appear different, given the range the system characteristics for radars used in the papers cited:*

- *Impulse 3 MHz system: 28 m range resolution in ice, 8 m in water (Welch et al., 2009) (Langley et al., 2011)*
- *UTIG/SOAR data, HiCARS 4MHz Bandwidth: 21 m range resolution in ice, 6.25 m in water (Carter et al., 2009)(Langley et al., 2011)*
- *HiCARS 14MHz Bandwidth: 6 m range resolution in ice, 1.8 m in water) (Wright et al., 2014)*

*A discussion of a damped signal from thin lakes or the effect of water depth more generally would be a useful addition somewhere in this paper. This feeds back into roughness interpretation, as a thin lake lid may look quite rough (as mentioned in the very last line of the conclusions).*

*R1: p.14/8:Here, in your discussion of sources of variance in reflection coefficient, you should introduce the idea of thin films / the effect of water depth.*

We agree that the effect of thin films on interpreting reflection coefficients is an area that deserves more study, particularly in the context of active lakes. However, we consider that to be out of scope for this paper -- our goal was to provide a thorough demonstration that traditional radar methods for detecting lakes do not provide a simple correspondence to observed surface features. We think that the dataset presented here provides a fantastic opportunity to continue the work of determining a reliable radar signature for an active lake that is known to be at a high stand.

As such, rather than introducing thin films in the Methods section, we have added a discussion of thin films in section 5.2 in the Discussion. See following comment for new text.

*R1: p.18/8-9: This statement is hard to defend -- can you imagine lifting the ice surface up 10 m through changes in water storage but not fully decoupling the ice from the rock to form a lake? You seem to argue this is analogous to what is observed at Thwaites, but I find that section of the text difficult to follow. If you want to keep this interpretation in, more discussion / justification is required.*

We agree that this paragraph was unclearly written; the re-written version also addressed the thin film analysis as requested in the previous comment. New text:

In interpreting reflection coefficients, there are a number of possibly-complicating factors. In an active lake system, it is likely that there are significant portions of the bed at the pressure melting point (water needs to be flowing into them), which would lead to lower contrast between the ice/water interface and the ice/bed interface. Depending on the depth of the lake, the roughness of the water/rock interface, and the salinity of its water, it is also possible that the radar return from the water/rock interface could interfere with that from the ice/water interface, lowering the observed reflection coefficient \cite{MacGregor2011}. \cite{Christianson2016} investigated anomalously low reflection coefficients in a region just offshore of the Whillans Ice Stream grounding zone, and concluded that they were due at least in part to sediments entrained in the ice not yet having melted out. Similarly, we could consider active lakes to be at one end of a continuum where stable radar lakes are the other end, and they are primarily differentiated by their water residence times.  The more rapidly evolving features may not have existed for long enough to melt a smooth roof, so we could be observing the preserved imprint of the bed at low stands or basal roughness advected from upstream of the lake. Supporting this view, some active lakes (Adventure Subglacial Trench) do appear on classic lake inventories, but they are typically the ones farther upstream, with longer cycle times.

Speculation is an appealing complement to reflection coefficient analysis for characterizing the basal interface. A classic lake would be expected to have a smooth, flat ice/water interface and appears as an isotropically specular surface, which requires decimeter scale smoothness over hundreds of meters. This concept has been used in earlier work to characterize the distribution of subglacial water: \citet{Young2016} looked at the anisotropy of individual lines by comparing the specularity of the first return to the amount of scattering recorded afterward, while \citet{Schroeder2013} leveraged a gridded survey of Thwaites Glacier. \cite{Schroeder2013} reported a pattern of anisotropic specularity in Thwaites Glacier, and concluded that it indicates ``canals'' of subglacial water pooling, aligned with the ice flow. Since canals require a sedimentary subglacial interface, this is consistent both with \citet{Carter2017}'s hypothesis that active subglacial lakes drain through canals and with \citet{Smith2017}'s observations of active lakes in the region of Thwaites Glacier where \citet{Schroeder2013} identified the water system transition.In the David Lakes region, we see an overall pattern of anisotropic specularity, including over the regions of surface elevation change, that is similar to that seen in Thwaites. Further work is needed to understand the overall hydrologic systems driving these active lakes, and the anisotropic specularity in this region provides an interesting constraint on possible organizations of water.

***R1: p.19/21-22: This point is a really interesting one, and I think should come up in the discussion earlier so the reader is prepared for it in the conclusions. Perhaps introduce this idea up near P.14, L,8.***

We have rewritten the relevant section in the discussion so that this point is made more clearly before the Conclusions. See previous comment for added text.

**Figures:**

***R2: Figures 1a/6/7: Add scale bar. Figure 6/7: Label lakes as D1/2.***

Done

***R1: Figure 4: Is the dz presented here normalized by the dt between observations? If not, it should be, so that you can reasonably compare observations that span 2003-2017 and those that span 2009-2017.***

We have fixed this figure to show normalized dz.

***R2: Figure 3: The radargram contrast is quite poor on paper and in print. Color range as mean +/- two standard deviations usually works well. Is the very top of the radargram indeed the surface, following elevation correction?***

We have improved the contrast and blanked out the potentially confusing above-surface radar signal.

***R2: Figure 8 caption: "we assume" not "we think"***

Changed

***R2: Figure 8: Narrow vertical scale (remove lower 20 dB).***

The lower 20dB were intentionally included in this figure to give room to show the interpretation of what the distribution would be if the noise threshold hadn't truncated it. We have trimmed 5dB from top/bottom and changed the aspect ratio to stretch the y-axis a little further.

***R2: Figure 9: Narrow the color scale used significantly. It's ok to saturate if less than / greater than symbols are used at the edge of the range. Otherwise, I'm not getting much out this reflectivity map.***

Color scale has been adjusted.

**Grammar and other Minor Edits:**

***R2: Throughout: CryoSat-2 not CryoSat.***

Done.

*R2: 2/7 to 3/4: Merge these two paragraphs.*

Done.

*R1: p8/13: The phrasing of this sentence is a bit awkward - perhaps: "Due to the existence of side lobes in the transmit/receive beam the first return criteria may underestimate ice thickness in rough terrain."*

Changed.

*R2: 9/3: "policy" is an awkward choice of terms here. Perhaps clarify what first-return picking means relative to other options?*

Changed wording policy->criteria, and clarified.

*R1: p11/7: The clause "cyclical active lakes are manifestations of feedback loops within this system" feels out of place, I suggest removing it.*

Removed.

*R2: 11/15-17: In the phrasing used here, "X%" makes more sense to describe the fraction of overburden pressure that earlier studies considered.*

Changed.

*R2: 12/10: Cite Matsuoka et al. (2012)*

Done.

*R2: 12/14: "Some studies attempt. . ." (some of these papers include some of the same authors even though they used different methods)*

Changed wording as requested.

*12/17: "David Glacier region"*

Changed.

*Editor: 12/24. ice loss or energy loss (?)*

Clarified: *dielectric* ice loss

*R2: 13/11-10 and 13/15: Description of graphical elements in figures (as opposed to what the displayed data mean for interpretation) should be reserved exclusively to figure captions.*

Removed.

**R2: 15/7: Because a graticule is consistently used in the figures (which I like), revise the phrasing "grid north" to (probably) "south". A north arrow in the figures could ameliorate the situation.**

Wording changed here and throughout.

**R2: 17/17: "a single snapshot seven years"**

Changed.

---

## Author Comment (AC2) · 30 Apr 2020

We thank the reviewers for their thorough, constructive, and encouraging feedback.

Our responses to both reviews are combined and have been uploaded as the supplement to AC1.
* * *

---

## Author Comment (AC3) · 30 Apr 2020

We thank the reviewers for their thorough, constructive, and encouraging feedback.

Our responses to both reviews are combined and have been uploaded as the supplement to AC1.